# Initially Reduced Linezolid Dosing Regimen to Prevent Thrombocytopenia in Hemodialysis Patients

**DOI:** 10.3390/antibiotics10050496

**Published:** 2021-04-26

**Authors:** Hitoshi Kawasuji, Yasuhiro Tsuji, Chika Ogami, Makito Kaneda, Yushi Murai, Kou Kimoto, Akitoshi Ueno, Yuki Miyajima, Yasutaka Fukui, Ippei Sakamaki, Yoshihiro Yamamoto

**Affiliations:** 1Department of Clinical Infectious Diseases, Toyama University Graduate School of Medicine and Pharmaceutical Sciences, 2630 Sugitani, Toyama 930-0194, Japan; kawasuji@med.u-toyama.ac.jp (H.K.); kane2214@med.u-toyama.ac.jp (M.K.); yushimurai@gmail.com (Y.M.); pyopyon16001@gmail.com (K.K.); aueno@med.u-toyama.ac.jp (A.U.); myuki@med.u-toyama.ac.jp (Y.M.); yasufuku@med.u-toyama.ac.jp (Y.F.); sakamaki@med.u-toyama.ac.jp (I.S.); 2Center for Pharmacist Education, School of Pharmacy, Nihon University, 7-7-1 Narashinodai, Chiba 274-8555, Japan; tsuji.yasuhiro@nihon-u.ac.jp; 3Department of Medical Pharmaceutics, Faculty of Pharmaceutical Sciences, University of Toyama, 2630 Sugitani, Toyama 930-0194, Japan; d1862303@ems.u-toyama.ac.jp

**Keywords:** linezolid, hemodialysis, thrombocytopenia, therapeutic drug monitoring, dosing regimen

## Abstract

This retrospective cohort study investigated the effects of an initially reduced linezolid dosing regimen in hemodialysis patients through therapeutic drug monitoring (TDM). Patients were divided into two groups depending on their initial dose of linezolid (standard dose of 600 mg every 12 h or initially reduced dose of 300 mg every 12 h/600 mg every 24 h). The cumulative incidence rates of thrombocytopenia and severe thrombocytopenia were compared between both groups using the Kaplan–Meier method and log-rank test. Eleven episodes of 8 chronic hemodialysis patients were included; 5 were in the initially reduced-dose group. Thrombocytopenia developed in 81.8% of patients. The cumulative incidence rates of thrombocytopenia and severe thrombocytopenia in the initially reduced-dose group were significantly lower than in the standard-dose group (*p* < 0.05). At the standard dose, the median linezolid trough concentration (C_min_) just before hemodialysis was 49.5 mg/L, and C_min_ at the reduced doses of 300 mg every 12 h and 600 mg every 24 h were 20.6 mg/L and 6.0 mg/L, respectively. All five episodes underwent TDM in the standard-dose group required dose reduction to 600 mg per day. Our findings indicate that initial dose reduction should be implemented to reduce the risk of linezolid-induced thrombocytopenia among hemodialysis patients.

## 1. Introduction

Thrombocytopenia is the most common adverse effect of linezolid treatment. Studies have shown that a linezolid trough concentration of >7–8 mg/L is associated with an increased risk of thrombocytopenia. This often limits its clinical use and may lead to discontinuation of treatment [1]. Furthermore, this adverse event can increase the risk of mortality among critically ill patients [2,3].

Linezolid dose adjustments for patients with end-stage renal disease (ESRD) are not currently required. Thus, it is easily prescribed for hemodialysis patients with Gram-positive infections [1]. However, the accumulating evidence indicates that the incidence of thrombocytopenia and subsequent linezolid discontinuation rates are significantly higher, and thrombocytopenia onset time is significantly shorter in patients with ESRD than those with normal renal function. This is caused by a systemic accumulation of linezolid in patients with ESRD [4,5,6,7,8].

Consequently, several groups have advocated for therapeutic drug monitoring (TDM) and dose adjustments to improve the safety and effectiveness of linezolid, especially among patients with renal impairment [9,10,11,12]. We recently demonstrated that TDM and TDM-guided dose modifications might be beneficial in preventing treatment failure of non-dialysis dependent patients with creatinine clearance (CL_CR_) < 60 mL/min [13]. In particular, we proposed that an empirical dose reduction to 300 mg every 12 h under TDM control may improve safety while maintaining appropriate effectiveness in these patients [13].

Very few studies have assessed the accumulation of linezolid that occurs in hemodialysis patients with repeated administration. Further, there is a need for dosing regimen optimization to ensure the safety of hemodialysis patients receiving linezolid.

Therefore, the current study aimed to evaluate the risk of linezolid-associated thrombocytopenia in hemodialysis patients and assess the effects of an initially reduced dosing regimen through the TDM of trough concentration.

## 2. Results

### 2.1. Patient Demographics and Clinical Characteristics

A total of nine patients (12 episodes) were included in the study. However, during one of the episodes, the patient had developed septic shock and was subsequently excluded from the study because of disseminated intravascular coagulation (DIC) and reliance on continuous hemodiafiltration during linezolid treatment. Hence, 11 episodes of 8 chronic hemodialysis patients were finally included. Five episodes were in the initially reduced-dose group (300 mg every 12 h or 600 mg every 24 h). There were no episodes co-administered with rifampicin, omeprazole, amlodipine, amiodarone, or dexamethasone. All episodes except for one underwent TDM during the treatment. The median (IQR) duration of linezolid treatment was 17.5 days (13–30 days) in the standard-dose group and 17 days (8.5–19 days) in the initially reduced-dose group. There were no significant differences between the standard and the initially reduced-dose groups regarding patient demographics and clinical characteristics (Table 1).

### 2.2. Frequency of Thrombocytopenia and Safety of the Initially Reduced Dosing Strategy

Thrombocytopenia developed in 81.8% of patients on linezolid therapy. In the standard-dose group, the median (IQR) time from initiation of linezolid to the occurrence of thrombocytopenia was 9 days (5–10.5 days) and 10 days (10–16 days) for the initially reduced-dose group. The standard-dose group showed a higher platelet count reduction rate relative to the initially reduced-dose group. There were no incidents of treatment failure (defined as toxicity or persistent infection) or reinfection after 30 days (Table 2).

Using Kaplan–Meier analysis, the cumulative incidence rates of thrombocytopenia and severe thrombocytopenia were significantly lower in the initially reduced-dose group than the standard-dose group (*p* = 0.023 and *p* = 0.036, log-rank test) (Figure 1).

All five episodes underwent TDM in the standard dose group required dose reduction to 600 mg per day (300 mg every 12 h or 600 mg every 24 h). They were implemented 3–6 days after the occurrence of thrombocytopenia, and the median (IQR) platelet count at the time of dose reduction was 110 × 10^3^/μL (104–335 × 10^3^/μL). Eight of nine episodes experienced thrombocytopenia in total were recovered from thrombocytopenia after the end of linezolid treatment. The remaining one episode in the standard dose group was recovered from thrombocytopenia at 11 days after TDM-based dose reduction during linezolid treatment.

### 2.3. Linezolid Trough Concentration at Standard and Reduced Doses

A total of 91 linezolid serum concentrations were measured. At the standard dose, the median (IQR) linezolid trough concentration (C_min_) just before hemodialysis (oral or intravenous route) was 49.5 mg/L (34.6–56.7 mg/L). The median (IQR) steady-state C_min_ at the reduced doses (300 mg every 12 h and 600 mg every 24 h) just before hemodialysis was 20.6 mg/L (19.5–26.3 mg/L) and 6.0 mg/L (3.9–16.0 mg/L), respectively (Figure 2).

The steady-state C_min_ at standard and reduced doses on the off-dialysis day could only be measured in one episode each (21.4 mg/L and 8.2 mg/L, respectively). The elimination efficiency of linezolid was found to be 26.4%, calculated using the 26 sets of consecutive concentrations measured just before and after intermittent hemodialysis.

There were three episodes where C_min_ was measured within 48 h of linezolid administration in the reduced-dose group. The values of C_min_ at 24 h were 8.3 mg/L and 7.6 mg/L, and the value at 48 h after linezolid administration was 5.5 mg/L.

## 3. Discussion

To the best of our knowledge, this is the first cohort study to assess the linezolid accumulation occurring with repeated standard and reduced dosing, its elimination efficiency during hemodialysis in a clinical setting, and the safety of the initially reduced dosing regimen under TDM control in hemodialysis patients. The standard-dose group exhibited a higher reduction rate of platelet count than the initially reduced-dose group. The cumulative incidence rates of thrombocytopenia and severe thrombocytopenia in the initially reduced-dose group were significantly lower than in the standard-dose group. There were no episodes of treatment failure (due to toxicity or persistent infection) or reinfection. All episodes that underwent TDM in the standard dose group required dose reduction to 600 mg per day (300 mg every 12 h or 600 mg every 24 h).

Hemodialysis is a significant means of linezolid elimination in patients with ESRD, as approximately 30% of the administered dose is removed during a 3 h hemodialysis session [14]. Brier et al. previously reported that the dose of linezolid does not need to be adjusted for hemodialysis patients. The recommendation was based on a study of single-dose administration during, but not after, hemodialysis among adults without infections [14]. This situation is clearly different from clinical settings, but due to the lack of evidence regarding the frequency of exposure-dependent adverse effects and linezolid concentrations on repeated standard dose administration, no specific indications for dose adjustments in hemodialysis patients have been provided.

Increasing evidence suggests that dialysis-dependent patients treated with the conventional dose of linezolid are 6–9 times more likely to experience hematological toxicity than patients with normal renal function. This is likely due to a systemic accumulation of linezolid [4,8,15]. In addition, the thrombocytopenia-associated linezolid discontinuation rate in hemodialysis patients was 62.5%, much higher than that among patients with normal renal function (2.3%) [8]. However, these previous studies did not assess linezolid concentrations and elimination efficiency during hemodialysis in a clinical setting [4,8,15]. In our study, there were no instances of linezolid discontinuation, likely due to the TDM-based dose adjustments that almost all patients received.

Previous works have demonstrated that TDM-guided dose adjustments maintaining a linezolid *C*_min_ range of 2–8 mg/L were beneficial for preventing treatment failure as well as for recovery from exposure-dependent thrombocytopenia while maintaining treatment efficacy [9,13]. We observed that TDM-based dose adjustments were also beneficial in hemodialysis patients, as they all required a reduced dose due to an extremely higher C_min_ at the standard dose. These observations were in line with a case–series study of peritoneal dialysis patients [7].

To prevent and reduce the risk of developing thrombocytopenia, early intervention may be key. Cojutti et al. reported that proactive TDM and platelet count assessment between days 3–5 of starting linezolid might be beneficial in preventing concentration-dependent thrombocytopenia [2]. Similarly, via classification and regression tree analysis, the predictive factors of linezolid-induced thrombocytopenia at the beginning of treatment for early interventions have been identified as a platelet count reduction to less than 2.3% from baseline at 96 h after the initial dose and a linezolid concentration greater than or equal to 13.5 mg/L at 96 h after the initial dose [16].

In the current study, severe thrombocytopenia was observed in 83.3% of the standard-dose group compared to 20.0% of the initially reduced-dose group. The cumulative incidence rates of developing thrombocytopenia and severe thrombocytopenia were significantly lower in the initially reduced-dose group than the standard-dose group. These are likely due to the early intervention via initial dose reduction to avoid linezolid overexposure, as patient demographics, baseline laboratory values, microorganisms, and the type of infection were not significantly different between the two groups. Other factors affecting the pharmacokinetics of linezolid include drug–drug interaction [17,18,19,20,21,22,23,24], liver dysfunction [25,26], and critical illness with/without acute kidney injury [27,28]. In our study, there were no episodes of linezolid co-administration with rifampicin, omeprazole, amlodipine, amiodarone, or dexamethasone, and there were no differences in the population of patients co-administered with levothyroxine in both groups, indicative of a limited effect of drug–drug interactions. In addition, there were no patients with liver dysfunction in our study.

Our findings showed that early intervention via initial dose reduction might prevent or delay the onset of thrombocytopenia and severe thrombocytopenia. On the other hand, there were no significant differences in the rates of thrombocytopenia and severe thrombocytopenia between the standard and the initially reduced-dose groups in the univariable analysis (Table 2). One of the reasons for this is that, despite using a reduced linezolid dose of 300 mg every 12 h/600 mg every 24 h, linezolid *C*_min_ within the optimal range was only seen in 30.8% (4/13), and exposure-dependent thrombocytopenia was eventually occurred even in the initially reduced dose group. Further reduction (for example, 300 mg per day) under TDM control may be needed in hemodialysis patients who require prolonged linezolid treatment.

There were no episodes of experiencing linezolid underexposure (<2 mg/L) even when administered a reduced dose of 600 mg per day (Figure 2). Based on these results, a reduced dose of 600 mg per day (300 mg every 12 h or 600 mg every 24 h) may be recommended as an optimal dose in hemodialysis patients. The analysis of the first *C*_min_ of the three episodes in the initially reduced-dose group suggested the possibility that hemodialysis patients may achieve optimal linezolid blood levels before reaching a steady state, even though their doses were initially reduced. However, the wide interindividual variability in C_min_ was observed and using TDM is thus essential to any intervention evaluating initial dose reduction in hemodialysis patients.

The major limitation of this study was its retrospective design. In addition, due to the small sample size, we were unable to perform population pharmacokinetics analysis, and it seems difficult to make a reliable conclusion about the availability of the initially reduced dosing regimen in hemodialysis patients based solely on our findings. Although we did not assess linezolid metabolites, a recent study reported that metabolite concentrations in paired samples were poorly correlated with linezolid concentrations (r^2^ = 0.26 for PNU-142300 and 0.06 for PNU-142586), and thus the relationship between these metabolites and thrombocytopenia remains unknown [29].

## 4. Materials and Methods

### 4.1. Study Design and Population

This is a retrospective cohort study investigating the frequency of linezolid-induced thrombocytopenia in hemodialysis patients with suspected or documented Gram-positive bacterial infections at Toyama University Hospital from April 2013 to December 2019. Patients ≥ 20 years of age, who were treated with oral or intravenous linezolid and had undergone hemodialysis at the beginning of antibiotic treatment, with adequate blood, wound, or urine specimens for microbiological culture, were included. Patients with DIC, bleeding complications, liver disease, or a total bilirubin level of more than twice the upper limit of normal ranges were excluded. Recurrent infection was considered a distinct episode only if it occurred more than one week after the initial episode and once antimicrobial therapy had been completed. Decisions regarding the modality and frequency of renal replacement therapy were made by the attending physician based on patient clinical characteristics. Most patients received hemodialysis three times per week.

### 4.2. Therapeutic Drug Monitoring Based on trough Concentration

Linezolid TDM was performed through an infectious disease (ID) consultation upon the request of the attending physicians responsible for patients, and the results were reported back to them. After the ID consultation at the start of linezolid treatment, an initial dose reduction to 600 mg per day (300 mg every 12 h or 600 mg every 24 h) was recommended for all hemodialysis patients, but the decision was left to the discretion of the attending physician. For one episode where TDM was not performed, linezolid C_min_ values could not be measured during treatment. This could have been due to a delay in ID consultation requests from the attending physician and/or difficulty in immediate measurements due to time constraints and limited human resources.

Serum concentrations were measured through peripheral venous blood sampling after starting linezolid therapy. These samples were collected just before and after hemodialysis on their on-dialysis day and before the next administration on an off-dialysis day. For patients on hemodialysis, the entire dose was administered once a day or split into two doses after dialysis therapy on their on-dialysis day. Concomitant administration of antimicrobial agents to treat infections was permitted. The times of linezolid administration (orally or intravenously) and blood collections were carefully checked, and samples deemed inappropriate were excluded from the analysis. Linezolid C_min_ was suitably measured, especially when ID physicians and/or attending physicians decided it necessary by reference to the course of platelet counts or C_min_ values throughout the treatment duration. In the event of thrombocytopenia with C_min_ > 10 mg/L, dose adjustment was recommended by ID physicians, targeting linezolid C_min_ within the optimal range of 2–8 mg/L [9,30].

### 4.3. Measurement of Linezolid Concentration

Steady-state serum C_min_ was defined as the concentration just before hemodialysis during on-dialysis days or just before the next administration on off-dialysis days ≥ 72 h after linezolid initiation or dose modification. The elimination efficiency of linezolid was calculated based on linezolid concentrations just before (C_min_) and after hemodialysis (C_HD_), as per the following equation:(1)$$\mathrm{Elimination}\;\mathrm{efficiency}\;\left(\%\right)=\frac{C_{min}\;-\;C_{HD}}{C_{min}}\;\times \;100$$

Serum linezolid concentrations were measured via high-performance liquid chromatography analysis, as described in the literature [11]. The intra- and inter-day coefficients of variation were always <5%, and the lower limit of detection was 0.1 mg/L.

### 4.4. Analysis Strategy

Patients were divided into two groups depending on their initial dose of linezolid administration. The first group received standard dosing (600 mg every 12 h), while the second group received reduced dosing (300 mg every 12 h or 600 mg every 24 h), which was initially administered under TDM.

### 4.5. Data Collection

Data were collected from the medical records of the study population. These include patient demographics, baseline laboratory and hematological parameters, source of infection, isolated microorganisms, linezolid dosage and serum C_min_ at each TDM, number of all instances of TDM, number of TDM under steady-state conditions, whether TDM for dosage adjustment was performed during linezolid treatment and whether it has resulted in any dose adjustments, treatment duration, as well as concomitant medications.

### 4.6. Evaluation of Myelosuppression

Thrombocytopenia and anemia were defined as unexplained reductions in platelet count and hemoglobin levels of >30% from patient baseline values before linezolid administration. Severe thrombocytopenia was defined as a 50% reduction from the baseline, and the nadir was defined as the lowest value during the study period. The lower range of normal platelet count was 158 × 10^3^/μL in our institution, and median (IQR) cutoff values of platelet count in cases of thrombocytopenia and severe thrombocytopenia were 142 (116–203) × 10^3^/μL and 108 (78–213) × 10^3^/μL, respectively. Recovery from thrombocytopenia was defined as the return and maintenance of platelet count values > 70% of baseline values after experiencing thrombocytopenia. Failure was defined as any discontinuation of linezolid therapy before the end of treatment, either because of toxicity or the persistence of infection. The reduction rate was calculated using the following equation:(2)$$\mathrm{Reduction}\;\mathrm{rate}\;(\%)=\frac{\mathrm{baseline}\;-\;\mathrm{nadir}}{\mathrm{baseline}\;}\times \;100$$

Complete blood counts and serum chemistry profiles were monitored two or three times per week, mainly just before hemodialysis, at the physician’s discretion. The incidence, onset time, and platelet reduction rate of thrombocytopenia were compared between the standard-dose group and the initially reduced-dose group.

### 4.7. Statistical Analysis

Categorical and continuous variables were compared using Fisher’s exact and the Mann–Whitney *U*-tests, respectively. The time from linezolid initiation to developing thrombocytopenia was estimated using the Kaplan–Meier method and log-rank test. A value of *p* ≤ 0.05 indicated statistical significance. All statistical analyses and plotting were performed using JMP Pro software, version 14.2.0 (SAS Institute, Cary, NC, USA).

## 5. Conclusions

In the current study, we found that an initial dose reduction to 600 mg per day (300 mg every 12 h or 600 mg every 24 h) under TDM control may be necessary to prevent and reduce the risk of concentration-dependent thrombocytopenia in hemodialysis patients. Prospective controlled trials involving larger numbers of patients are warranted to confirm our findings and suggestions on using linezolid in hemodialysis patients, and specific studies in peritoneal dialysis patients are also required.

## Figures and Tables

**Figure 1 antibiotics-10-00496-f001:**
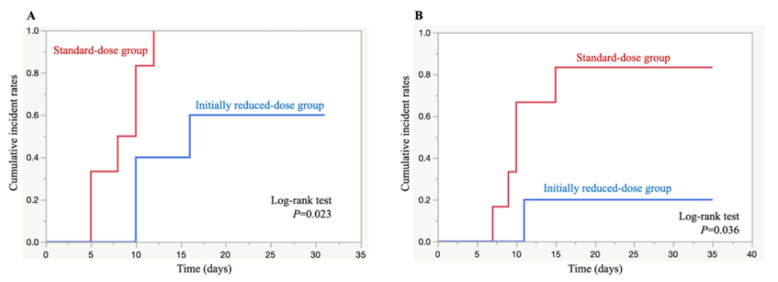
Kaplan–Meier curves of thrombocytopenia (**A**) and severe thrombocytopenia (**B**) development time after the initiation of linezolid therapy in the standard-dose group (red) and initially reduced-dose group (blue).

**Figure 2 antibiotics-10-00496-f002:**
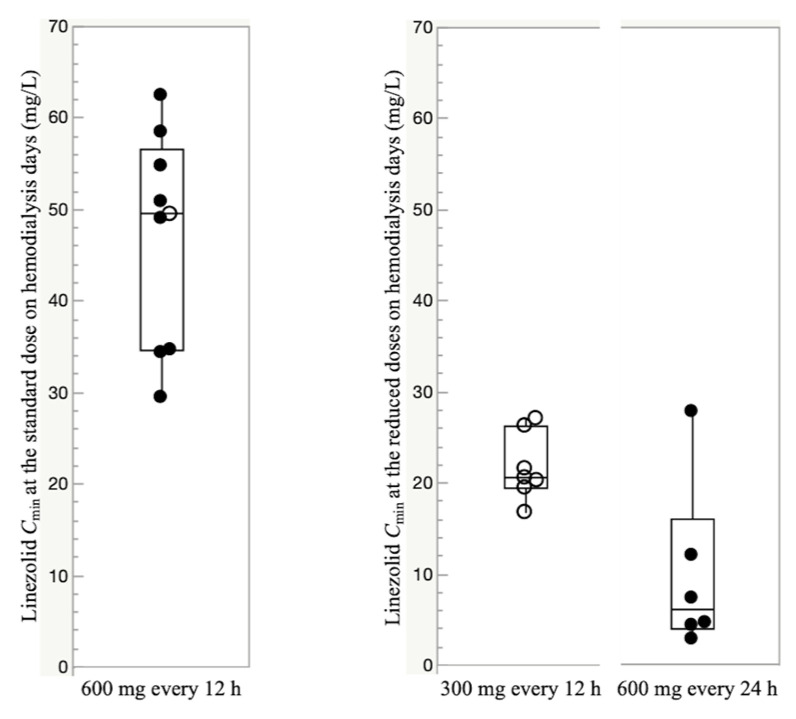
Boxplots of trough concentration (C_min_) at the standard dose and the reduced doses to 300 mg every 12 h and 600 mg every 24 h on hemodialysis days. For each boxplot, the horizontal line across the box represents the median, each box represents the range between the 25th and 75th percentiles, the two whiskers represent the minimum and maximum values within 1.5 × IQR, and points beyond the whiskers represent outliers. Closed circles represent C_min_ administered through the intravenous route, and open circles represent C_min_ administered through the oral route.

**Table 1 antibiotics-10-00496-t001:** Characteristics of episodes in the standard-dose group and initially reduced-dose group.

Characteristics	All, *n* = 11	Standard Dose Group, *n* = 6 (54.5%)	Initially Reduced-Dose Group, *n* = 5 (45.5%)	*p*-Value
Demographics				
Age (years), median (IQR)	56 (41–60)	55 (42–64)	59 (41–76)	0.78
Sex (male/female), (%/%)	8/3 (72.7/27.3)	4/2 (66.7/33.3)	4/1 (80.0/20.0)	1.00
Body weight (kg), median (IQR)	56.7 (46.3–64.0)	53.2 (42.3–78.3)	56.7 (42.8–60.4)	0.93
Body mass index (kg/m^2^), median (IQR)	19.8 (17.1–22.6)	21.2 (16.1–26.1)	19.8 (17.7–21.1)	0.65
Laboratory, median (IQR)				
Serum creatinine (mg/dL)	6.8 (5.9–9.8)	6.8 (5.6–8.5)	8.5 (6.1–10.6)	0.31
eGFR	6.1 (4.5–8.5)	6.8 (5.7–8.3)	4.5 (4.2–9.3)	0.65
Total bilirubin (mg/dL)	0.2 (0.2–0.4)	0.3 (0.2–0.4)	0.2 (0.2–0.4)	1.00
Baseline hematological parameters				
Hemoglobin concentration (g/dL), median (IQR)	9.8 (8.5–11.6)	9.7 (8.2–10.3)	8.8 (7.3–10.4)	0.78
Platelet count (×10^3^/μL), median (IQR)	243 (177–319)	240 (184–488)	180 (166–201)	0.12
Main reason for linezolid				
Type of infection, *n* (%)				
Skin and soft tissue infections, and surgical site infections	8 (72.7)	3 (50.0)	5 (100.0)	0.18
Mediastinitis	3 (27.3)	2 (33.3)	1 (20.0)	1.00
Bone and joint infections	2 (18.2)	1 (16.7)	1 (20.0)	1.00
Respiratory tract infections	1 (9.1)	1 (16.7)	0 (0.0)	1.00
Microbiological isolate, *n* (%)				
MRSA	5 (45.5)	3 (50.0)	2 (40.0)	1.00
MR-CoNS	3 (27.3)	2 (33.3)	1 (20.0)	0.81
No isolate, unknown	3 (27.3)	1 (16.7)	2 (40.0)	0.55
Linezolid dosage and exposure				
Empirical/target therapy, *n*/*n* (%/%)	3/8 (27.3/72.7)	0/6 (0.0/100.0)	2/5 (40.0/60.0)	0.061
Dose (mg/kg/day), median (IQR)	-	22.7 (15.5–28.6)	10.6 (10.0–15.3)	0.022
Number of all TDM instances, median (IQR)	10 (3–12)	10.5 (2.8–14.5)	5 (2.5–11.5)	0.52
Episodes with TDM assessment performed during linezolid treatment, until end of treatment	10 (90.9)	5 (83.3)	5 (100.0)	1.00
Duration of linezolid treatment (days), median (IQR)	17 (13–21)	17.5 (13–30)	17 (8.5–19)	0.31
Cotreatment, *n* (%)				
Levothyroxine	6 (54.5)	2 (33.3)	4 (80.0)	0.24
Other antimicrobials, *n* (%)				
Meropenem	2 (18.2)	1 (16.7)	1 (20.0)	1.00
Piperacillin/tazobactam	2 (18.2)	2 (33.3)	0 (0.0)	0.45

Abbreviations: eGFR, estimated glomerular filtration rate; MR-CoNS, methicillin-resistant coagulase-negative staphylococci; TDM, therapeutic drug monitoring.

**Table 2 antibiotics-10-00496-t002:** Linezolid-related adverse events and clinical outcome in the standard-dose group and initially reduced-dose group.

Variables	All, *n* = 11	Standard-Dose Group, *n* = 6 (54.5%)	Initially Reduced-Dose Group, *n* = 5 (45.5%)	*p*-Value
Type of toxicity, *n* (%)				
Thrombocytopenia	9 (81.8)	6 (100.0)	3 (60.0)	0.18
Median time from initiation of therapy to development of thrombocytopenia (*n* = 9), median days (IQR)	10 (6.5–11)	9 (5–10.5)	10 (10–16)	0.18
Severe thrombocytopenia	6 (54.5)	5 (83.3)	1 (20.0)	0.080
Nadir platelet count (×10^3^/μL), median (range)	97 (54–208)	81.5 (57–208)	131 (54–180)	0.65
Reduction rate of platelet count (%), median (IQR)	57.8 (35.5–67.0)	63.1 (52.7–76.1)	35.5 (6.4–54.7)	0.055
Anemia	7 (63.6)	5 (83.3)	2 (40.0)	0.24
Gastrointestinal intolerance	2 (18.2)	2 (33.3)	0 (0.0)	0.45
Hyponatremia	2 (18.2)	2 (33.3)	0 (0.0)	0.45
Clinical outcome, *n* (%)				
Failure	0 (0.0)	0 (0.0)	0 (0.0)	-
Thirty-day reinfection	0 (0.0)	0 (0.0)	0 (0.0)	-

## Data Availability

The datasets used and analyzed during the current study are available from the corresponding author on reasonable request.

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
