# Peer review of "Initially Reduced Linezolid Dosing Regimen to Prevent Thrombocytopenia in Hemodialysis Patients"

_antibiotics, 2021, doi:10.3390/antibiotics10050496_

Round 1

Reviewer 1 Report

In this interesting , clear and well written article, authors present a retrospective comparative study of usage of linezolid in hemodialysis comparing standard doses to reduced doses (together with drug dosage) and its influence on the occurrence of thrombocytopenia. They show a very significant reduction of thrombocytopenia and and smaller platelet count decrease in those patients with reduced dose of linezolin. Their work is very welcome since publications on that topic are scarce with a poor clinical translation.

Some comments and suggestions to improve the manuscript to the standards of the Journal :

Table 2 : authors should give the median nadir of the platelet count in each group (with their range).

Figure 1 : I suggest to the authors to duplicate this figure in 1a as actually and 1 b showing the Kaplan-Meyer curves for severe thrombocytopenia (together with the result of the log rank-test).

In their conclusion authors should indicate that  prospective controlled trials are warranted to confirm their findings and suggestions on the use of linezolin in hemodialysis patients. They should also indicate that specific studies are required in PD patients.

Methods -paragraph 4.6 Authors, please give the lower range of normal platelet count (in healthy Japanase population) as the absolute level of platelet count in case of thrombocytopenia and severe thrombocytopenia 

Reviewer 2 Report

  1. Currently, manuscript structure is introduction, results, discussion, and methods. Please check if journal format requires manuscript structure – introduction, methods, results, discussion.
  2. This is cohort study, not case-control study
  3. The sample of this study is small although investigators found statistical difference between the risk of thrombocytopenia between standard and reduced dose. It is unlikely to make reliable conclusion based on sample size of 11
  4. Could you explain why there is no significant difference in rate of thrombocytopenia in table 2 (p=0.18) but there is significant difference in Kaplan-Meier analysis (p=0.023)
  5. Did patients need dose reduction or discontinuation of linezolid when thrombocytopenia occurs. Is there any improvement in platelet count after dose reduction or discontinuation of linezolid?
  6. How often was CBC (hemoglobin, WBC, and platelet count) monitored while patients are on linezolid

Round 2

Reviewer 2 Report

all of my comments have been addressed